# Private and interpretable clinical prediction with quantum-inspired tensor train models

## Abstract

Publicly available clinical machine learning models pose an underappreciated privacy risk: their parameters or outputs can be exploited to recover information from patients whose data were used during training. Moreover, this risk is exacerbated by models such as logistic regression (LR), which are typically preferred in clinical settings for their transparency. To assess this empirically, we attack LORIS, a publicly available LR model for immunotherapy response prediction hosted on a U.S. government website. From evaluations through its public interface, we recover the model parameters and identify the training cohort with high confidence. More broadly, we design cohort-level membership inference attacks under three levels of adversarial access—binary black-box, continuous black-box, and white-box—and apply them to both LR models and shallow neural networks (NNs) trained on the same task. Our results reveal that even a cohort of 35 patients can be reliably identified within training sets of hundreds to thousands, and that common practices such as cross-validation amplify rather than mitigate this risk. To address these vulnerabilities, we propose a quantum-inspired defense based on tensorizing discretized models into tensor trains (TTs). This representation obfuscates model parameters and preserves accuracy, while offering black-box protection comparable to practical Differential Privacy baselines. Additionally, the TT representations retain LR interpretability and extend it through efficient computation of marginal and conditional distributions, enabling this richer analysis also for black-box models such as NNs. Our results establish tensorization as a practical, post-hoc tool for private, interpretable, and effective clinical prediction.

## 1 Introduction

Machine learning (ML) is increasingly used for clinical prediction but poses critical privacy risks, as models trained on sensitive medical data can inadvertently leak individual information (Fredrikson et al., 2014; Sweeney, 2015). In domains where interpretability is essential, such as clinical prediction, intuitive models like logistic regression (LR) are often preferred, yet they are particularly vulnerable to such attacks. More complex models like neural networks (NNs) are harder to attack, but their complexity also makes it challenging to design strong, accuracy-preserving defenses, leaving them vulnerable.

In this work, we study these vulnerabilities in a relevant real-world setting. We design a cohort-level membership inference attack in which an adversary attempts to determine which patient cohorts were included in the training of a given model. Although this task is easier than individual membership inference—i.e., identifying single patients from the training dataset—in medical settings each public cohort may correspond to a small group of patients collected from a specific hospital or study, whose identification may still reveal sensitive information.

To defend against such attacks while preserving the key benefits of clinical prediction models, we propose a defense based on quantum-inspired tensor network (TN) models, focusing on tensor trains (TTs). Specifically, we learn TT models via TT-RSS (Pareja Monturiol et al., 2025) from model outputs discretized into $b$ bins, thereby compressing the output information available to a black-box adversary while removing parameter-level information not present at the black-box level. The use of discretized outputs is motivated by prior

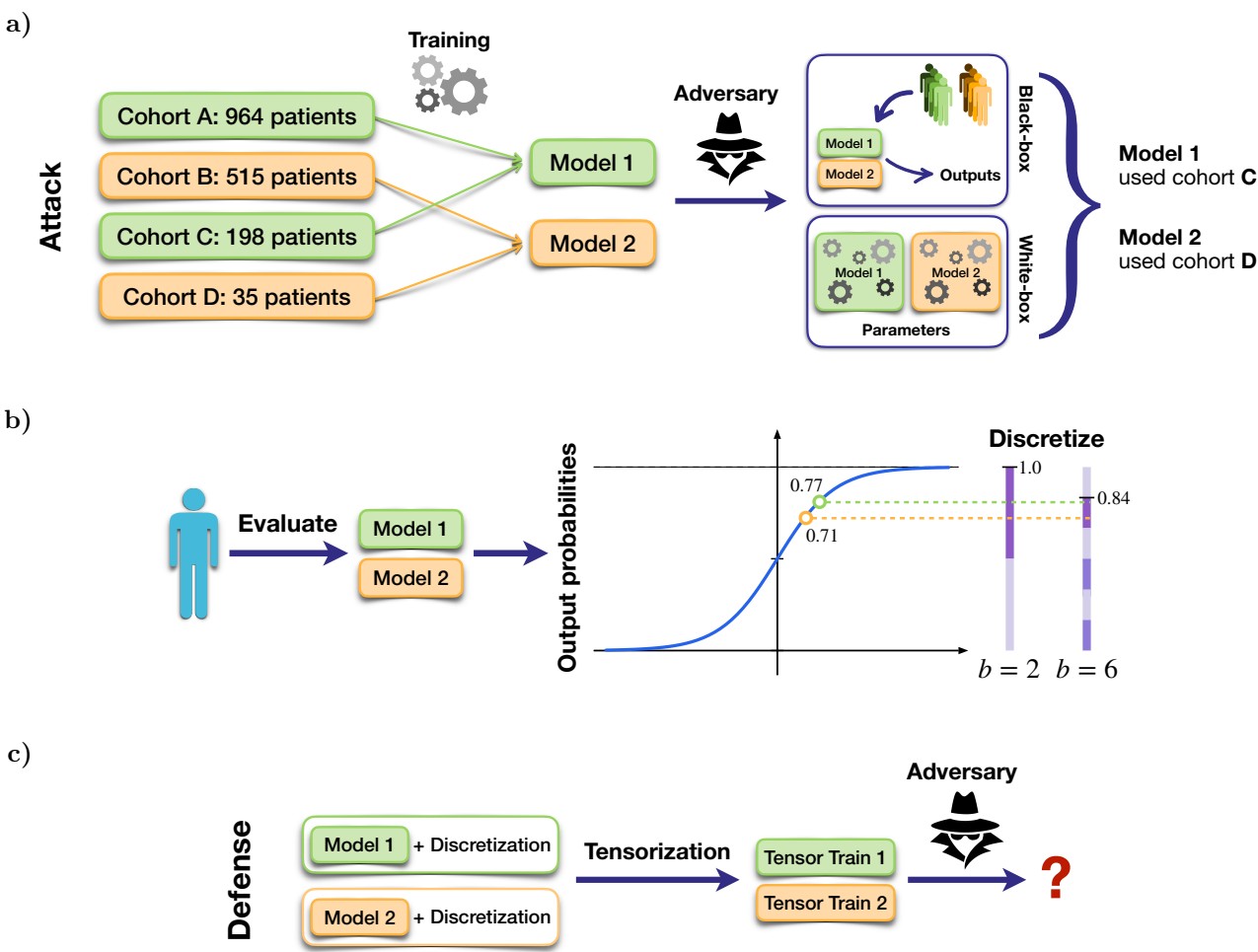

Figure 1: **Overview of the attack and defense setting. (a)** Membership inference attack. Given a set of public patient cohorts, an adversary attempts to determine which cohorts were used to train a target model, exploiting either its output probabilities (black-box access) or its parameters (white-box access). **(b)** Output discretization. Continuous model output probabilities are discretized into $b$ bins (here $b = 2$ and $b = 6$), compressing the output space and reducing the training-data information available to a black-box adversary. **(c)** Tensorization defense. Discretized model evaluations are used to learn a tensor train representation via TT-RSS (Pareja Monturiol et al., 2025), enhancing black-box privacy and removing parameter-level information not present at the black-box level, thus protecting also against white-box attacks.

work showing that compressing a model's output space can reduce membership inference risk (Jia et al., 2019; Yang et al., 2020; 2023; Ye et al., 2022). Although we do not provide formal black-box privacy guarantees, discretization alone, especially in the extreme case $b = 2$, removes all output information beyond the binary classification behavior needed to preserve predictive accuracy. The obfuscation of parameter-level information is motivated by formal white-box privacy guarantees for TNs (Pozas-Kerstjens et al., 2024). These formal white-box guarantees apply to the resulting TT representation, ensuring that access to its parameters reveals no information beyond its black-box behavior. Our solution therefore combines output-space compression with parameter-level obfuscation, while retaining predictive accuracy and interpretability. Figure 1 illustrates the full pipeline.

As a case study, we attack LORIS (Chang et al., 2024), a publicly available LR model for prediction of response to immune checkpoint blockade (ICB) immunotherapy. As shown by Chang et al. (2024), LORIS outperforms all other studied models—including NNs—in terms of accuracy and generalization, while also

providing direct access to feature importance and producing output scores that grow monotonically with true population level response probability. For these reasons, LORIS was deployed through a public web interface hosted on a U.S. government website for informational purposes[1]. Although our study leverages the availability of open-source code and data for LORIS to validate results, we introduce uncertainty regarding the information available to an adversary, allowing the framework to generalize to more restricted settings.

Under this threat model, we design a membership inference attack under binary black-box (bBB), continuous black-box (cBB), and white-box (WB) access, using a shadow model approach that trains multiple models with varied hyperparameters and datasets, followed by an adversarial meta-classifier to predict which public cohorts were included in the training set. To further demonstrate the generality of our tensorization approach, we perform analogous experiments on shallow NNs trained with the same data and objectives. The TT-RSS approach is architecture-agnostic and can be similarly applied to NNs, as it only requires access to model evaluations to construct the TT representation, with its applicability mainly limited by the class of functions that can be efficiently represented by a TT model. Additionally, we compare the results with Differential Privacy (DP) defenses for both LR and NN models. This empirical comparison is not intended to imply equivalence between the approaches, but rather to contextualize the observed privacy protection of our method alongside practical DP baselines against membership inference attacks.

Our results show that unprotected models leak substantial training-cohort information, that averaged LR models obtained by repeated cross-validation—a common practice used to obtain the final LORIS model—are more vulnerable than vanilla LRs despite similar predictive performance, and that even the inclusion of a 35-patient cohort can be detected under sufficiently informative access. Regarding our proposed defense, we show that tensorizing discretized models degrades attack performance across all access levels, reducing WB attacks applied to the TT parameters to random guessing while providing BB protection comparable to DP and maintaining predictive accuracy close to the unprotected models. We also find that the discretization parameter $b$ provides some control over privacy protection by restricting the amount of output information retained, analogous in spirit to the role of $\epsilon$ in DP (Dwork, 2006a). This suggests potential future work on deriving theoretical bounds on the information recoverable for a given value of $b$, or on designing alternative output obfuscation techniques with formal DP guarantees. Additionally, TT approximations preserve key properties of LORIS, such as response monotonicity, while enhancing interpretability through efficient computation of marginals and conditionals. This supports feature-sensitivity analysis and enables the construction of cancer-type-specific models without retraining. Importantly, the same techniques can be directly applied to tensorized NNs, providing interpretability for these black-box models.

The remainder of this paper is structured as follows. Section 2 reviews related work and preliminaries on privacy attacks, defenses, and TT models. Section 3 presents the attack setting, target models, tensorization defense, and privacy results. Section 4 analyzes the interpretability of TT models. Finally, Section 5 discusses conclusions, limitations, and future directions.

## 2 Related work and preliminaries

The widespread adoption of ML systems increases the risk of leaking sensitive personal data. Prior work has extensively examined these vulnerabilities and proposed various defenses.

### 2.1 Privacy attacks

A wide range of attacks exploit privacy vulnerabilities in ML, leveraging either black-box or white-box access. Key examples include model inversion (Fredrikson et al., 2014), model classification (Ateniese et al., 2015), and membership inference (Shokri et al., 2017), which vary in scope from extracting individual training samples to uncovering global data patterns. Importantly, the level of adversarial access to a model has a substantial impact on the amount of information that can be recovered, with recent work showing that access to model parameters can enable full reconstruction of training samples (Balle et al., 2022; Haim et al., 2022; Oz et al., 2024).

---

[1]LORIS is available at: `https://loris.ccr.cancer.gov/`.

In this work, we adopt the membership inference approach to identify groups of samples present in the training set. More specifically, our attack identifies which public patient cohorts were used for training. This differs from standard individual membership inference, but remains privacy-relevant in clinical settings, where a cohort may correspond to a small study, a rare cancer subtype, or a specific institution. This group-level attack is particularly relevant in our case study of LORIS, because it is trained on a collection of datasets from different sources.

LR models are particularly exposed in this setting. For LR, reconstruction attacks can yield closed-form solutions (Balle et al., 2022), underscoring how widely used, transparent models can be the most exposed. This motivates our focus on LORIS and on LR-based clinical predictors, while also evaluating whether similar risks extend to shallow NNs trained on the same task.

## 2.2 Defense mechanisms

Given the diversity of privacy-related attacks, various defense mechanisms have been proposed. Among these, Differential Privacy (Dwork, 2006b) stands out for its rigorous framework. DP quantifies the likelihood that an attacker can infer whether a specific user's data was included in a statistical process. A randomized algorithm $\mathcal{A}$ is $\varepsilon$-DP if, for any set of outcomes $\mathcal{S}$ in the range of $\mathcal{A}$, it satisfies

$$\log \left( \frac{\mathrm{P}[\mathcal{A}(D) \in \mathcal{S}]}{\mathrm{P}[\mathcal{A}(D') \in \mathcal{S}]} \right) \leq \varepsilon, \tag{1}$$

where $D$ and $D'$ differ by a single element. This metric guides the addition of calibrated noise to achieve a target $\varepsilon$, based on the sensitivity of the function being protected (Dwork, 2006a; Dwork & Roth, 2014). For LR models, common defenses add noise either to the objective or to the final parameters (Chaudhuri et al., 2011); for NNs, the standard approach is DP-SGD (Abadi et al., 2016), which adds noise to gradients at each training step. However, the noise required for meaningful privacy guarantees ($\varepsilon \ll 1$) often degrades performance and may exacerbate group disparities (Bagdasaryan et al., 2019; Hansen et al., 2024). Hence, there is no consensus on how to set $\varepsilon$ in a practically meaningful way (Garfinkel et al., 2018): while small values are theoretically ideal, larger values may still prevent attacks in practice without significantly harming accuracy (Ziller et al., 2024).

Beyond DP, recent work has explored whether standard ML practices can improve privacy. Pruning introduces small errors that resemble DP-like protection (Huang et al., 2020), while knowledge transfer reduces dependence on specific training data (Shejwalkar & Houmansadr, 2020). Several works show that non-private models produce overly spread output scores, and that compressing this space, e.g., by injecting crafted noise to prevent membership inference, improves privacy (Jia et al., 2019; Yang et al., 2020; 2023). With a similar goal, other approaches add noise to the output scores, providing DP guarantees with minimal utility loss (Ye et al., 2022; Papernot et al., 2017).

Our approach builds on these ideas: tensorization acts as a knowledge-distillation mechanism that converts a model into an efficient, interpretable representation with obfuscated parameters, retaining only BB information (Pozas-Kerstjens et al., 2024). To further enhance BB privacy, we apply tensorization to discretized scores that collapse the model's output space into a smaller subdomain, while preserving the necessary information for classification. The tensorization mechanism itself is independent of the obfuscation method, so alternative noise-based output-obfuscation approaches could similarly be applied before tensorizing to enforce DP. Thus, our approach may resemble work on private low-rank approximation, such as Kapralov & Talwar (2013), but at the level of full-model decomposition, in contrast with techniques that use low-rankness for private fine-tuning (Liu et al., 2025).

## 2.3 Tensor train models

Tensor networks are low-rank decompositions of high-dimensional tensors with roots in quantum many-body physics. They offer compact, interpretable representations of quantum states (Pérez-García et al., 2007; Orús, 2014; Cirac et al., 2021) and have recently been adapted to machine learning (Stoudenmire & Schwab,

2016; Novikov et al., 2018). TNs have been applied to model compression (Novikov et al., 2015; Tomut et al., 2024), explainable AI (Tangpanitanon et al., 2022; Aizpurua et al., 2024), anomaly detection (Wang et al., 2020), and robustness (Mossi et al., 2025). Importantly, TNs offer formal WB privacy guarantees: due to the explicit characterization of their gauge freedom, it is possible to move across the multiple parameterizations that represent the same model and to define a canonical parameterization that effectively obfuscates all information beyond its BB behavior (Pozas-Kerstjens et al., 2024). These guarantees therefore reduce WB access to BB access, although information recoverable from model evaluations may still be extracted through the TT parameters and exploited in privacy attacks, as we discuss in the next section for the particular case of LR models.

Throughout this work, we focus on one-dimensional TNs known as tensor trains (Oseledets, 2011). An order-$N$ tensor $T \in \mathbb{R}^{d^N}$ admits a TT representation with *ranks $r_n$* if it can be written as

$$T(i_1, \ldots, i_N) = G_1(i_1) \cdots G_N(i_N), \tag{2}$$

where the *cores $G_n$* are $r_{n-1} \times r_n$ matrices and $r_0 = r_N = 1$. This structure also supports continuous functions of the form

$$f(x_1, \ldots, x_N) = \sum_{i_1, \ldots, i_N} \mathrm{W}(i_1, \ldots, i_N) \, \phi_1(i_1, x_1) \cdots \phi_N(i_N, x_N), \tag{3}$$

where W is a TT-format coefficient tensor and $\phi_n(i_n, x_n)$ are vector-valued embedding functions indexed by $i_n$. To ensure non-negative probability scores, it is standard to define distributions via the Born rule: $p(x) = |f(x)|^2$. Further details on TTs, including efficient marginalization and conditioning, are provided in Appendix B.

TTs can be trained using SGD or physics-inspired variants (Stoudenmire & Schwab, 2016). Alternatively, TT representations can be constructed via low-rank decompositions, bypassing high-dimensional optimization. Recent techniques based on sketching (Hur et al., 2023) and cross interpolation (Núñez Fernández et al., 2025) achieve this using only function evaluations—i.e., BB access—to approximate continuous functions in TT form. A recent method, TT-RSS, extends this idea to tensorize pre-trained NNs using a small evaluation dataset (Pareja Monturiol et al., 2025). This results in an efficient procedure, requiring $O(|D|^2 Nd)$ model evaluations on a dataset of *pivots D* and an additional $O(|D|^3 Nd)$ to assemble the TT. Since TT-RSS relies on randomized linear algebra, the stability of the resulting decomposition depends on the choice and size of the pivot dataset $D$. A detailed performance analysis is provided in Pareja Monturiol et al. (2025), where it is shown empirically that choosing $D$ as a small subset of the training data is sufficient to obtain reliable decompositions. In this work, we adopt TT-RSS to tensorize models.

## 3 Privacy analysis

To evaluate the privacy risks of clinical prediction models and compare defense strategies, we design a membership inference attack based on shadow-model training. Assuming an adversary with access to multiple public datasets, the attack determines which of them were used to train a model under varying levels of access. In this section, we first describe the experimental setting and methodology, following the protocol used to evaluate LORIS and the proposed defenses, and then present the privacy results.

### 3.1 Setting and overview

The attack considered in this study aims to identify which data cohorts, from a set of publicly accessible cohorts, are included in the training set of a given clinical prediction model. To this end, we follow a shadow-model-based approach, in which multiple models are trained under different configurations, with training sets constructed from possible combinations of cohorts. We then train a multi-label meta-classifier to predict the presence or absence of each cohort in the training set, leveraging either the model outputs (bBB or cBB access) or information contained in the model parameters (WB access). We next specify the

clinical datasets, target and protected models, attack construction, evaluation metrics, and implementation details used in this analysis.

### 3.1.1 Clinical datasets

In our study, we evaluate privacy vulnerabilities within the setting of Chang et al. (2024). The underlying task in that work is the prediction of treatment response in cancer patients receiving immune checkpoint blockade (ICB) immunotherapy, using tumor mutational burden (TMB) together with additional clinical, genomic, and pathological variables. Although TMB has been proposed as a biomarker of ICB efficacy, it is not universally predictive, motivating the development of multivariate models that combine TMB with other patient characteristics. Importantly, all patient cohorts considered originate from studies that pursue this same task of predicting binary ICB response.

The datasets used in this setting consist of multiple patient cohorts spanning 18 solid tumor types and up to 18 features, including tumor information, standard clinical variables, and blood-based markers. The represented cancer types include: non-small cell lung (NSCLC), renal, melanoma, head and neck, bladder, sarcoma, gastric, central nervous system (CNS), colorectal, endometrial, hepatobiliary, cervical (CLC), esophageal, pancreatic, mesothelioma, ovarian, breast, and cancers of unknown primary.

To establish a common modeling framework across heterogeneous cohorts, Chang et al. (2024) proposed a six-feature logistic regression model—the LORIS score—based on TMB, Patient's Systemic Therapy History (PSTH), Albumin, Neutrophil-to-Lymphocyte Ratio (NLR), Age, and Cancer Type. Table 1 summarizes the cohorts and their main characteristics.

Table 1: Summary of the main dataset characteristics, including cohort size, cancer types, number of features provided, and original references.

| Dataset | Size | Cancer types | #Features | Reference |
|---------|------|--------------|-----------|-----------|
| Cho1 | 964 | 16 solid tumors | 18 | Chowell et al. (2022) |
| Cho2 | 515 | | | |
| MSK1 | 453 | 15 solid tumors | 13 | Chang et al. (2024) |
| MSK2 | 104 | CNS / Unkown primary | 12 | |
| Shim | 198 | NSCLC | 13 | Shim et al. (2020) |
| Kato | 35 | 8 rare tumors | 6 | Kato et al. (2020) |

### 3.1.2 Target models and defense mechanisms

As target models, we consider LRs and NNs. Following Chang et al. (2024), we train *averaged* LRs via 20 repetitions of 3-fold cross-validation. While LORIS used larger numbers of repetitions and folds, we found this configuration sufficient to obtain comparable results. For comparison, we also train *vanilla* LRs through a single training run on an 80% split of the corresponding dataset. In both cases, the hyperparameters are solver = "saga", penalty = "elasticnet", class_weight = "balanced", max_iter = 100, l1_ratio $\in \{0, 0.5, 1\}$, and C $\in \{0.1, 1, 10\}$. For NNs, we train 2-layer MLP classifiers following the same procedure as the vanilla LRs, adopting the best hyperparameters reported by Chang et al. (2024): two hidden layers of size 19, binary cross-entropy loss, and Adam optimization for 100 epochs with batch_size = 32, lr = $10^{-3}$, and weight_decay = $10^{-5}$.

For each trained LR and NN, we build a TT model via the TT-RSS tensorization algorithm (Pareja Monturiol et al., 2025), using 50 random samples from the corresponding training set as pivots. Model evaluations on these pivots are discretized into $b$ bins, with $b \in \{2, 6, 10\}$: values $< 0.5$ map to the lower bin limit and values $> 0.5$ to the upper, preserving the property that output probabilities sum to 1. The resulting TTs have $N = 22$ cores (including one for the output), ranks $r = 2$, input dimension $d = 2$, and use polynomial embeddings $\phi(x) = [1, x]$. To accommodate the higher complexity of NNs, we use 80 pivots and ranks $r = 5$. After tensorization, the TT cores are randomized via a gauge transformation, fully obfuscating the

parameters and preventing any leakage under WB access beyond the information already recoverable through BB access. Further details on the TT structure and efficient computations are provided in Appendix B.

It is important to note that only the outputs of the original LR or NN used during tensorization are discretized according to $b$, constraining the amount of information from the original model retained in the TT representation. The resulting TT nevertheless remains a continuous function: cBB attacks and AUC evaluation use its continuous output scores, whereas bBB attacks use only its final binary classifications. Thus, $b$ is a parameter of the tensorization procedure, whereas bBB and cBB describe the information available to the attacker after the model has been constructed.

Finally, for comparison with a standard privatization approach, we also train DP models (LR-DP, NN-DP) from scratch. Since DP training of LR is restricted to solver = "lbfgs" and penalty = "l2", we fix max_iter = 100 and vary the privacy budget $\varepsilon \in \{0.1, 1, 10, 100\}$, where $\varepsilon = 100$ nearly matches the non-DP case. Only vanilla models are considered, as averaging would cancel the injected noise and effectively increase $\varepsilon$. For NNs, we follow the same training setup as in the non-DP case but apply DP-SGD with max_grad_norm = 1, $\delta = 10^{-4}$, and $\sigma \in \{20, 5, 1, 0\}$, which correspond approximately to privacy budgets $\varepsilon \in \{0.2, 1, 10, \infty\}$. To achieve these budgets, we reduce the number of epochs to 50. Because the available DP implementations require different optimization settings, these experiments compare complete practical defense pipelines rather than providing a controlled isolation of the effect of DP noise alone.

### 3.1.3 Membership inference attack design

We assume the adversary knows the model architecture and training procedure, up to uncertainty in the specific hyperparameter configurations used. The adversary also has access to the public cohorts $\{C_1, \ldots, C_M\}$ and to models trained on datasets $D$ formed as unions of these cohorts, and has sufficient resources to train shadow models and meta-classifiers. The attack therefore evaluates membership among a known set of candidate cohorts and does not address discovery of arbitrary unknown cohorts.

We distinguish between three levels of adversarial access: binary black-box (bBB), from binary model classifications only; continuous black-box (cBB), from continuous output probabilities; and white-box (WB), from model parameters. The attack proceeds by constructing a dataset of shadow models—LRs, NNs, and TTs as described above—each trained under different hyperparameter configurations and training sets. From each model, we collect the available model information together with the corresponding cohorts used for training, forming the input to a multi-label classifier that learns to identify the presence of each cohort in the training set. Formally, the attack consists of the following steps:

1. For each hyperparameter configuration and for each possible combination of cohorts in $\mathcal{C}$ forming a dataset $D$, train 100 shadow models.

2. Build a dataset of (model information, membership label) pairs, where model information is the available representation under the chosen access level—bBB, cBB, or WB—and the membership label is a binary vector whose $m$-th entry is 1 if cohort $C_m \subset D$ and 0 otherwise.

3. Train an adversarial classifier minimizing independent cross-entropy losses for each $C_m$, yielding a meta-model that, given model information as input, returns a vector where entry $m$ gives the probability that $C_m \subset D$.

As adversarial classifiers, we use MLP multi-label classifiers with three hidden layers of sizes 32, 16, and 8, and an output layer of size 6 (one per public cohort). The input size depends on the access type. For BB attacks, each shadow model is evaluated on 100 samples—the same samples for all models—drawn randomly from the union of all cohorts; the resulting vector of continuous or binary outputs serves as input to the adversary. For WB attacks we collect full model parameters: for LR, 22 parameters (21 coefficients + intercept); for NN, all per-layer parameters concatenated into an 818-dimensional vector; and for TT, all $N = 22$ cores concatenated into a single vector of 168 (TT-LR) or 1 020 (TT-NN) dimensions. All parameters are rescaled when needed to operate on raw inputs (see Appendix C). The adversary MLPs are trained with activation = "relu", solver = "adam" and max_iter = 100. Since WB attacks exhibited greater variability,

predictions are averaged across 5-fold cross-validation. To obtain robust statistics, this procedure is repeated five times for both WB and BB attacks, and the Hamming scores reported in Tables 3 and 4 correspond to the mean across these five repetitions. The standard deviations across these runs are generally of order $10^{-3}$–$10^{-2}$ and do not exhibit systematic differences across the evaluated settings; therefore, we omit them from the tables for readability.

Due to the monotonicity of LR, model parameters can be exactly recovered from sufficiently precise scores and suitable queries (see Appendix D), making cBB and WB access equivalent, although WB is typically easier to exploit. Since tensorization approximates LR outputs with a TT representation, it is also possible to recover LR coefficients from TT evaluations, with greater accuracy as $b$ increases, making the approximation more precise. Thus, because TT parameters are fully obfuscated, we assume a WB attacker would instead reconstruct the original LR coefficients and attack those directly; accordingly, we report also these attacks in Tables 3 and 4. This strategy, however, assumes that the adversary knows that the TT approximates an underlying LR model, which may be a strong assumption, and is not readily extensible to other settings, such as NNs. We hypothesize that output-obfuscation strategies with formal DP guarantees, applied before tensorization instead of discretization, could provide stronger protection, though we leave this direction for future work.

### 3.1.4 Evaluation metrics

Attack performance is reported as the Hamming score: the proportion of correctly predicted cohort-membership labels across all public patient cohorts and shadow-model instances. A score of 0.5 corresponds to chance-level prediction (random guessing), and a score of 1.0 indicates perfect identification of training datasets.

Clinical model performance is reported as median balanced accuracy and AUC across repeated shadow-model runs. Balanced accuracy uses a threshold based on Youden's J statistic rather than the standard 0.5 threshold, as the latter produced irregular results for DP models under strong noise. Tensorization occasionally produces degenerate models with accuracies near 50%; although rare, these can distort mean values, motivating the use of medians. This behavior may be caused by the intrinsic randomness of TT-RSS and can be mitigated by increasing the pivot dataset size, as shown in the performance analysis of Pareja Monturiol et al. (2025). However, because the computational cost of tensorization grows with the dataset size, we use relatively small pivot sets to limit the computational burden.

### 3.1.5 Implementation

All experiments[2] were run on an Intel Xeon CPU E5-2620 v4 with 256 GB RAM and an NVIDIA GeForce RTX 3090, using Scikit-Learn for LR models and NN-based attacks (Pedregosa et al., 2011), Diffprivlib for DP variants (Dwork, 2006b), PyTorch for NN models (Paszke et al., 2019), Opacus for DP-SGD training (Yousefpour et al., 2022), and TensorKrowch for TT models (Pareja Monturiol et al., 2024).

### 3.2 Experimental results

### 3.2.1 Public models leak patient cohort information

We begin by attacking LORIS as deployed. Since WB attacks are typically the strongest, we apply them to two sets of coefficients: (i) the LR coefficients of LORIS as released by Chang et al. (2024), and (ii) coefficients reconstructed from the web interface. Although the interface returns rounded probabilities rather than exact scores, we approximately invert the monotonic mapping defined for LORIS (Fig. 2c) to obtain usable coefficients (see Appendix D).

Table 2 shows that Cho1 is correctly identified as the training dataset in both cases, consistent with Chang et al. (2024), which reports Cho1 as the training set and Cho2 as the test set. The recovered coefficients are noisier and assign high probability to Cho2 as well; however, Cho1 remains the dominant prediction. Since Cho1 and Cho2 correspond to train/test splits of the same dataset, both drawn from the same patient cohort,

---

[2]The code is publicly available at: `https://anonymous.4open.science/r/tts4privacy`.

this spurious assignment likely reflects shared data characteristics. These results demonstrate that, even with noisy reconstructed coefficients, an adversary can infer training data membership with high confidence, highlighting the privacy risks of releasing or exposing LR parameters.

Table 2: WB attack scores for LORIS, using (i) the released model parameters from Chang et al. (2024), and (ii) coefficients reconstructed from queries to the web interface. Each score reflects the probability assigned by the meta-classifier to the presence of the corresponding cohort in the training set, where a score of 1.0 indicates identification with high confidence. Note that Cho1 and Cho2 correspond to train/test partitions of the same original dataset, and thus high scores for both are expected.

|  | Cho1 | Cho2 | MSK1 | MSK2 | Shim | Kato |
|---|---|---|---|---|---|---|
| Released | **1.0000** | 0.0007 | 0.0001 | 0.0000 | 0.0003 | 0.0000 |
| Reconstructed | **0.9944** | 0.8138 | 0.0440 | 0.0173 | 0.0005 | 0.0007 |

### 3.2.2 Privacy risk generalizes across model types and access levels

Having established the vulnerability of LORIS, we assess whether this risk generalizes to identifying other cohorts, including smaller ones, and to more complex models such as NNs. We also compare DP-based and tensorization-based defenses against such attacks. Table 3 reports Hamming scores across all model types and access levels, together with model performance metrics to illustrate the effect of the evaluated defense mechanisms on utility. Specifically, for each patient cohort, AUC scores are reported as the median over all models whose training set includes that cohort; these therefore reflect training-set performance rather than generalization, for which we refer the reader to Appendix A.1. Additionally, Appendix A.2 reports Hamming scores for the identification of each cohort separately, illustrating the generalizability of attacks across cohorts.

In Table 3 we distinguish between *vanilla* LR models, trained on a single run, and *averaged* LR models, trained via repeated cross-validation with coefficients averaged across folds—the procedure used to train LORIS. Notably, averaged LR models are more vulnerable than vanilla ones despite similar predictive performance. The variance reduction from cross-validation mitigates sample bias but also amplifies differences across models, making averaged models more identifiable than vanilla ones—a counterintuitive finding with direct implications for clinical practice, where cross-validated averaged models are often recommended precisely to improve generalization. However, ensemble methods may still be privacy-preserving when the individual models or the aggregation method are themselves private, as in PATE (Papernot et al., 2017).

Beyond the cross-validation finding, the results yield three main observations. First, unprotected LR and NN models yield the highest attack scores, underscoring their vulnerability when released without protection. Second, larger $\varepsilon$ and $b$ values correspond to higher data leakage, paired with higher performance, as expected. Third, attack scores increase with deeper levels of access, with cBB and WB achieving high values in many cases. Somewhat unexpectedly, WB attacks on NNs achieve lower scores than BB attacks, likely reflecting the difficulty of extracting structured information from more complex parameter spaces.

### 3.2.3 Tensorization compared with Differential Privacy

Although DP provides protection at all access levels, it typically comes at a high cost in performance. The lowest privacy budgets ($\varepsilon \approx 0.1, 1$), arguably the only ones offering strong rigorous guarantees, have the largest impact on model utility, making them impractical. For the largest values ($\varepsilon \approx 100, \infty$), AUC scores approach non-DP levels, while attacks still perform worse than in the unprotected case, indicating that even negligible noise helps in practice. For NN-DP models, predictive performance is hindered even at $\varepsilon = \infty$, reflecting differences in the DP-SGD pipeline, such as the reduced number of epochs or gradient clipping, rather than an effect of noise addition. Among the evaluated DP configurations, the intermediate case $\varepsilon \approx 10$ offers the best empirical balance between privacy and utility.

For TT models, the most restrictive setting, $b = 2$, yields attack scores comparable to those of the DP baselines with $\varepsilon \in (1, 10)$, while maintaining high AUC scores. Although larger $b$ values slightly improve

Table 3: Hamming scores and median AUC scores for all model types and access levels. Hamming scores reflect the proportion of correct cohort-membership predictions across all cohorts and attacked models; a score of 0.5 corresponds to random guessing and 1.0 to perfect identification. Standard deviations are generally of order $10^{-3}$–$10^{-2}$ and do not exhibit systematic differences across settings; hence, we omit them for readability. AUC scores are reported as the median over all models whose training set contains that cohort, and therefore reflect training-set performance rather than generalization. Vanilla LR models are trained on a single 80/20 split, while averaged LR models are trained via repeated cross-validation with coefficients averaged across folds, following the procedure used to train LORIS (Chang et al., 2024). *WB attacks on TT-LR use LR coefficients reconstructed from TT evaluations rather than TT parameters directly (see Appendix D).

| | | Attack Hamming score | | | Model AUC | | | | | |
| | | bBB | cBB | WB | Cho1 | Cho2 | MSK1 | MSK2 | Shim | Kato |
|---|---|---|---|---|---|---|---|---|---|---|
| LR | (vanilla) | 0.8178 | 0.9129 | 0.9330 | 0.74 | 0.76 | 0.71 | 0.62 | 0.61 | 0.75 |
| | (averaged) | **0.9149** | **0.9910** | **0.9999** | 0.74 | 0.77 | 0.71 | 0.63 | 0.61 | 0.75 |
| LR-DP | ($\varepsilon = 0.1$) | 0.5314 | 0.5352 | 0.5088 | 0.51 | 0.51 | 0.50 | 0.50 | 0.50 | 0.50 |
| | ($\varepsilon = 1$) | 0.5710 | 0.5808 | 0.5178 | 0.58 | 0.58 | 0.55 | 0.53 | 0.54 | 0.53 |
| | ($\varepsilon = 10$) | 0.7163 | 0.7840 | 0.6403 | 0.73 | 0.75 | 0.69 | 0.62 | 0.60 | 0.67 |
| | ($\varepsilon = 100$) | 0.7663 | 0.8610 | 0.8672 | 0.74 | 0.76 | 0.70 | 0.63 | 0.61 | 0.74 |
| TT-LR | ($b = 2$) | 0.6666 | 0.8231 | 0.5117 (0.7461$^\star$) | 0.68 | 0.70 | 0.67 | 0.59 | 0.61 | 0.62 |
| | ($b = 6$) | 0.7535 | 0.8604 | 0.5112 (0.7979$^\star$) | 0.71 | 0.73 | 0.69 | 0.62 | 0.61 | 0.66 |
| | ($b = 10$) | 0.7687 | 0.8710 | 0.5104 (0.8129$^\star$) | 0.71 | 0.74 | 0.69 | 0.62 | 0.61 | 0.67 |
| NN | | **0.7375** | **0.8608** | **0.6336** | 0.77 | 0.79 | 0.71 | 0.63 | 0.64 | 0.79 |
| NN-DP | ($\varepsilon \approx 0.2$) | 0.5222 | 0.6186 | 0.5125 | 0.59 | 0.59 | 0.49 | 0.60 | 0.53 | 0.54 |
| | ($\varepsilon \approx 1$) | 0.5085 | 0.6420 | 0.5033 | 0.63 | 0.62 | 0.53 | 0.62 | 0.55 | 0.55 |
| | ($\varepsilon \approx 10$) | 0.5924 | 0.6659 | 0.5008 | 0.66 | 0.64 | 0.58 | 0.63 | 0.57 | 0.57 |
| | ($\varepsilon = \infty$) | 0.6198 | 0.7370 | 0.6360 | 0.74 | 0.74 | 0.67 | 0.64 | 0.62 | 0.72 |
| TT-NN | ($b = 2$) | 0.5759 | 0.6184 | 0.5061 | 0.68 | 0.68 | 0.62 | 0.61 | 0.58 | 0.60 |
| | ($b = 6$) | 0.6252 | 0.8267 | 0.5018 | 0.72 | 0.75 | 0.68 | 0.62 | 0.63 | 0.71 |
| | ($b = 10$) | 0.6544 | 0.8215 | 0.5025 | 0.73 | 0.75 | 0.68 | 0.62 | 0.63 | 0.71 |

performance, the gains are modest compared to the sharp increase in attack success. This illustrates an advantage of tensorization as a knowledge-distillation mechanism: it reconstructs an effective model from highly restricted information while preserving utility. Although our proposed defense does not provide rigorous BB privacy guarantees, the $b = 2$ setting can be viewed as a soft, empirical upper bound on the information recoverable while preserving most of the original classification behavior, with tensorization errors potentially reducing both attack and predictive accuracy relative to this idealized limit. In contrast, DP methods inject noise directly into the learning process to make models trained on neighboring datasets indistinguishable, without explicitly preserving utility. Consequently, noise injection can have a disparate impact on model accuracy (Bagdasaryan et al., 2019). Although we do not observe this effect in our experiments, it may become more pronounced in higher-dimensional settings.

Regarding WB access, direct attacks on the gauge-randomized TT parameters yield accuracies close to 50%. For TT-LR models specifically, since they approximate the original LR, we additionally apply the coefficient reconstruction technique used in Section 3.2.1, yielding the starred WB scores in Table 3. These attacks do not outperform BB attacks on the original LR, consistently with the fact that the TT is constructed solely from $b$-discretized LR evaluations. This attack is specific to the LR format and requires the attacker to know the underlying architecture. For TT-NN models, where NN parameters cannot be recovered, direct attacks on the TT parameters remain close to chance.

### 3.2.4 Even small cohorts of 35 patients can be identified

As an illustrative case, we consider the extreme task of distinguishing models trained only on Cho1 (964 samples) from those trained on Cho1 plus the small Kato cohort (35 samples). This simulates a high-risk scenario where an adversary detects the inclusion of a very small subgroup, bringing the setting closer to individual membership inference than our broader cohort-level experiments. Table 4 shows attack accuracies for the Kato label. As expected, bBB attacks are nearly random. In contrast, averaged LRs reach approximately 92% detection under cBB and achieve perfect classification under WB. Notably, even vanilla LRs under WB access attain approximately 71% accuracy. These results show that even a 35-sample cohort can be reliably identified within a larger dataset. Model averaging and WB access amplify leakage, while TT models remain robust and do not reveal the presence of Kato under any access type.

For context, Appendix A.3 reports model performance on Kato. TT models show some degradation, especially in AUC, but this does not fully explain the attack results: the performance gap between training on Cho1 or Cho1+Kato is similar for TTs and LRs, and NNs achieve even higher accuracies while leaking no information.

Table 4: Hamming scores of adversarial classifiers distinguishing models trained on Cho1 from those trained on Cho1+Kato, evaluated on the Kato label. Kato is the smallest cohort in the study, with only 35 patients. Each score reflects the probability assigned by the meta-classifier to the presence of Kato in the training set. $^\star$WB attacks on TT-LR use LR coefficients reconstructed from TT evaluations (see Appendix D).

|  |  | bBB | cBB | WB |
|---|---|---|---|---|
| LR | (vanilla) | 0.5981 | 0.6217 | 0.7141 |
|  | (averaged) | **0.5065** | **0.9182** | **1.0000** |
| TT-LR | $(b = 2)$ | 0.5375 | 0.5811 | 0.4966 (0.5521$^\star$) |
| NN |  | 0.5253 | 0.5403 | 0.5246 |
| TT-NN | $(b = 2)$ | 0.4779 | 0.5226 | 0.5152 |

## 4 Interpretability with tensor trains

Beyond privacy protection, interpretability is essential in clinical prediction. The utility of LORIS lies not only in its accuracy, but also in its ability to provide insights into relevant features and produce scores monotonically correlated with population level response probability—properties that help verify whether the model leverages known biological processes, contributing to its trustworthiness. Here we show that TT models retain similar interpretability, leveraging efficient computation of marginal and conditional distributions.

### 4.1 Feature sensitivity

The interpretability of LR models comes from their coefficients, which quantify how each feature affects the log-odds of the predicted outcome. For TT models, an analogous notion is obtained from conditional and marginal distributions. Unlike LR coefficients, which isolate each feature's independent effect, TT sensitivities may vary with other features due to non-linearity. To emulate LR coefficients, we marginalize over all but one feature and the response, and measure how the predicted score changes under a unit increment of the selected feature. This procedure yields independent sensitivity scores that can be computed efficiently within the TT structure (see Appendix B).

To evaluate this approach, we tensorized a vanilla LR trained on Cho1 and compared TT sensitivity scores with LR coefficients. We also included a tensorized NN model to assess how NN-based sensitivities compare to LR insights. As shown in Fig. 2a, LR and TT-LR scores align almost perfectly after normalization by the maximum absolute value to remove scale differences. TT-NN yields similar relative patterns, albeit with larger scaling differences. These results confirm that TTs preserve LR interpretability while extending the framework to more complex black-box models like NNs.

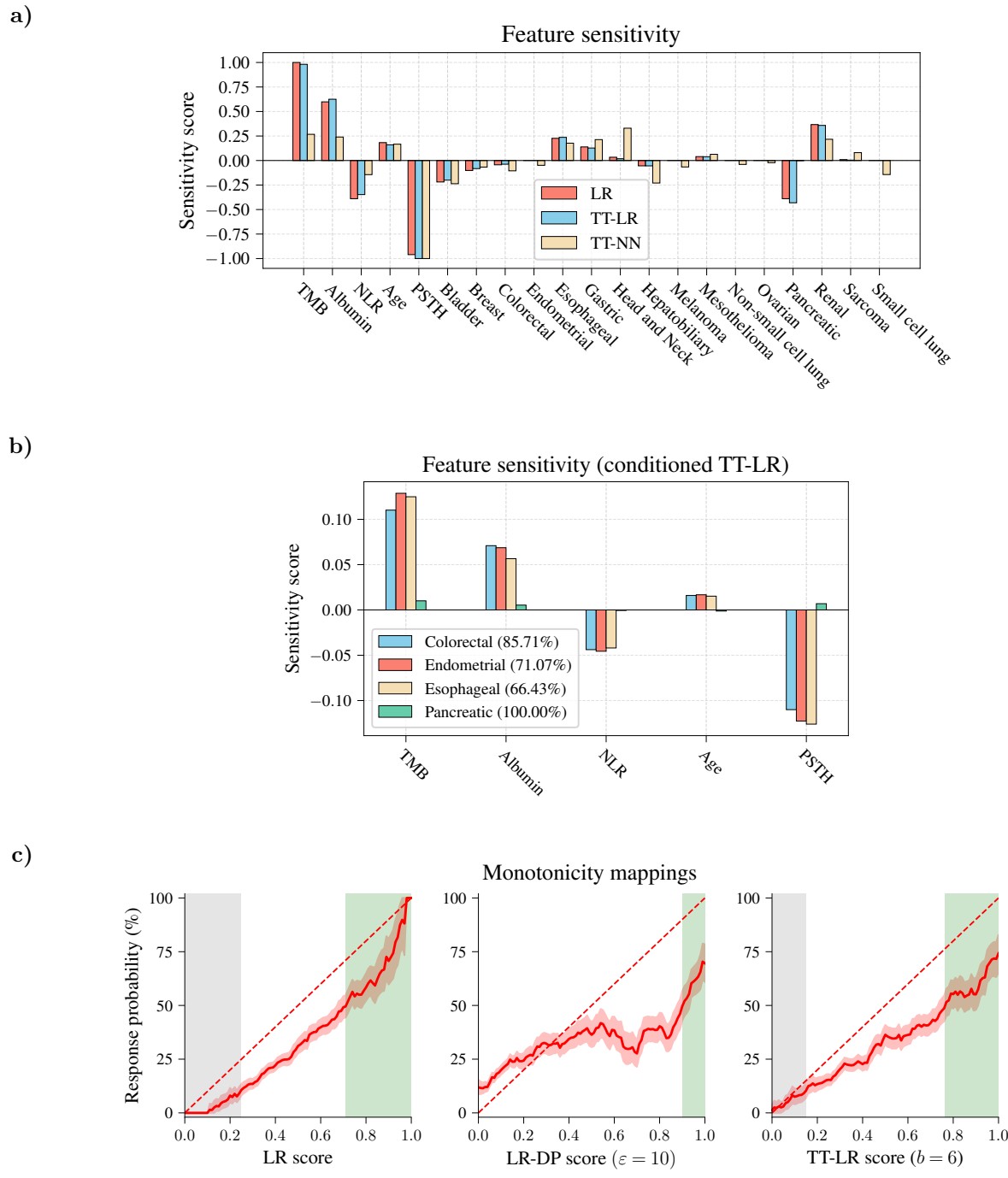

Figure 2: **Interpretability of TT models. (a)** Feature sensitivity scores from LR and TT models (TT-LR and TT-NN, $b = 6$). LR scores correspond to model coefficients, while TT scores are obtained via marginalization. All values are normalized by the maximum absolute score. **(b)** Feature sensitivities from cancer-type-conditioned TT-LR models ($b = 6$), obtained without retraining. The legend indicates cancer type and balanced accuracy of each conditioned model on the corresponding data. **(c)** Monotonicity plots of LR, LR-DP ($\varepsilon = 10$) and TT-LR ($b = 6$) scores with respect to true response probability, estimated via bootstrapping (95% confidence intervals). Shaded regions indicate participants with unlikely (gray, $< 10\%$) or likely (green, $> 50\%$) response probability. Limits of these regions from left to right: (0.25, 0.71), (0.00, 0.90), and (0.15, 0.76).

## 4.2 Feature sensitivity by cancer type

TTs also allow conditional analysis, enabling sensitivity computation for specific subgroups. Conditioning on cancer type produces smaller TT models that capture type-specific behaviors (see Appendix B). Unlike the normalized comparison above, scores are directly comparable across cancer types since they are computed with the same method.

Figure 2b shows feature sensitivities for colorectal, endometrial, esophageal, and pancreatic cancers. While LR would provide identical scores across types, TTs reveal subtle variations. In particular, pancreatic cancer yields uniformly small sensitivities. This occurs because all pancreatic cancer patients in Cho1 are non-responders: the model achieves 100% accuracy simply by assigning very low response probabilities to all samples, independently of their features. Consequently, no feature appears relevant for prediction within this subgroup. These results highlight how TT interpretability can reveal subgroup-specific effects not captured by linear models.

## 4.3 Monotonicity of TT scores

A key property of LORIS scores, highlighted by Chang et al. (2024), is their monotonic relation with response probability: higher scores directly correlate with a greater chance of response, allowing clinicians to use the score as an intuitive ranking of treatment suitability. Although LR models are trained on binary labels, their scores align with mean response probabilities across patients sharing a given score. We verify this via bootstrapping to compute 95% confidence intervals for a vanilla LR model trained on Cho1. For comparison, we construct the same mapping for a DP-protected LR model, with $\varepsilon = 10$, and for a tensorized LR model with $b = 6$.

Figure 2c shows the results. For the tensorized model, we observe a clear monotonic trend with a lower slope than the unprotected model, reflecting that discretization pushes the model toward more extreme values. Increasing the number of bins improves the approximation, yielding a mapping closer to that of the LR model, though with potentially weaker privacy protection. In contrast, the DP model exhibits a noisier, non-monotonic mapping as a result of the noise injection mechanism used during training. This highlights that, although DP can provide privacy protection in practice, its effects may be less predictable and less uniform across settings.

## 5 Discussion

Our results show that publicly available clinical ML models pose a concrete and immediate privacy risk. LORIS, a model developed with rigorous scientific standards and made available to support reproducibility, can be attacked to identify training cohorts with high confidence from its published parameters or web interface alone. This reflects a gap in current practice: privacy evaluation is not yet a standard component of clinical model deployment. Achieving high predictive performance is critical for clinical models, and must be balanced with interpretability; privatization mechanisms that compromise either are unlikely to be adopted in practice. However, as we show in this work, certain procedures can be avoided or implemented to meaningfully improve privacy with little impact on performance.

A particularly actionable finding concerns cross-validation. Although useful for model selection, averaging LR models for deployment should be avoided, as it amplifies privacy risks without offering meaningful accuracy gains. This result is counterintuitive: practitioners who use repeated cross-validation to obtain stable, well-calibrated models are inadvertently making those models easier to attack. The mechanism is clear in retrospect—variance reduction sharpens the fingerprint that training data leave on model parameters—but it is not widely recognized in the clinical ML community. Ensemble methods remain viable, but privacy-preserving aggregation strategies such as PATE (Papernot et al., 2017), which adds noise to the output aggregation step to ensure DP, should be considered.

Tensorization addresses these vulnerabilities without requiring changes to the training pipeline. Its key practical advantage over DP is that it is post-hoc: it can be applied to existing models—including LORIS—requiring only black-box access, without retraining or access to the original patient data, with its applicability

mainly limited by the expressiveness of the TT representation. Black-box privacy arises from tensorizing discretized rather than raw continuous scores, with the discretization level $b$ providing an empirical privacy–utility knob, while white-box privacy follows from the freedom in TT parameterizations (Pozas-Kerstjens et al., 2024).

Comparing TT-based protection with DP, we find similar privacy and utility for certain combinations of $b$ and $\varepsilon$, although a direct correspondence is not possible without DP-style output perturbation. In some evaluated settings—most notably for NNs—even binary discretization yields attack scores comparable to small-$\varepsilon$ DP baselines while retaining higher predictive accuracy. For LR models, the relative behavior depends more strongly on the access level and DP configuration, but tensorization still provides a competitive empirical privacy–utility trade-off. Furthermore, while DP provides formal theoretical guarantees, selecting a privacy budget that provides useful predictive performance typically requires empirical evaluation and, in training-based implementations, repeated retraining. Moreover, our results confirm prior findings (Ziller et al., 2024): in our experiments, only relatively large $\varepsilon$ values preserve practical accuracy, while smaller values providing stronger guarantees severely degrade performance.

These conclusions should be interpreted within the scope of our empirical threat model: we evaluate cohort-level inference over a known set of public cohorts, and the BB protection of tensorization is measured against the considered attacks rather than established as a formal privacy guarantee.

A natural extension of this work is to combine tensorization with other output obfuscation methods (Jia et al., 2019; Yang et al., 2020; Ye et al., 2022), and in particular with DP-style output perturbation. This could combine the rigorous privacy guarantees of $\varepsilon$-DP with the post-hoc nature and potential utility advantages of tensorization, while preserving the parameter-level obfuscation of the resulting TT representation.

Beyond privacy, we showed that TTs recover LR interpretability while enabling richer analyses, including subgroup-specific effects. More importantly, TT interpretability extends naturally to tensorized NNs, suggesting that our approach can help "open the box" of otherwise opaque models. Although our analysis of TT-NNs is relatively limited, recent work suggests promising directions for mechanistic interpretability with TN models (Pearce et al., 2025). Thus, even when privacy is not the primary goal, tensorization provides a powerful framework for extracting insights from pre-trained models, reinforcing its value as a broadly applicable tool for both privacy and interpretability.

The tensorization framework is not limited to immunotherapy response prediction or to a specific model architecture: since TT-RSS only requires model evaluations, it can in principle be applied post-hoc to a broad range of pre-trained clinical ML models, provided that they admit an efficient TT approximation. Looking forward, our results motivate further investigation of tensorization as a standard post-hoc step before clinical model deployment, analogous to data anonymization before publication.

## Broader impact statement

This work identifies privacy risks in publicly available clinical prediction models and evaluates a post-hoc defense intended to reduce training-data leakage while preserving predictive accuracy and interpretability. A potential negative impact is that the described attacks could facilitate attempts to infer the participation of small cohorts in deployed models. We mitigate this risk by restricting the analysis to publicly released model interfaces, parameters, source code, and cohorts, without accessing or reconstructing any non-public patient data.

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

# A  Additional privacy results

## A.1  Performance of models trained on Cho1

In this section we provide additional results supporting the conclusions of the main text. Specifically, we report: (i) performance metrics of models trained on Cho1 (the largest cohort, 964 samples) and evaluated on all cohorts; (ii) per-cohort attack accuracies for LR, NN, and TT models; and (iii) performance of models trained on Cho1 versus Cho1+Kato.

Figure 3 shows the overall performance of models trained exclusively on Cho1, reporting balanced accuracy distributions across all public cohorts. Tensorization occasionally produces degenerate models with accuracies near 50%, which, although rare, can distort mean values. This behavior may arise from the intrinsic randomness of TT-RSS and can be mitigated by increasing the pivot dataset size, as shown in the performance analysis of Pareja Monturiol et al. (2025). For this reason, we report median accuracies and AUC scores throughout, as they better capture typical behavior. Since the remaining distributions are approximately Gaussian and symmetric, median and mean coincide, making median values representative. The right panel of Figure 3 further shows how the performance of DP models improves with increasing $\varepsilon$, as the added noise decreases and the distribution converges to the narrow non-DP case.

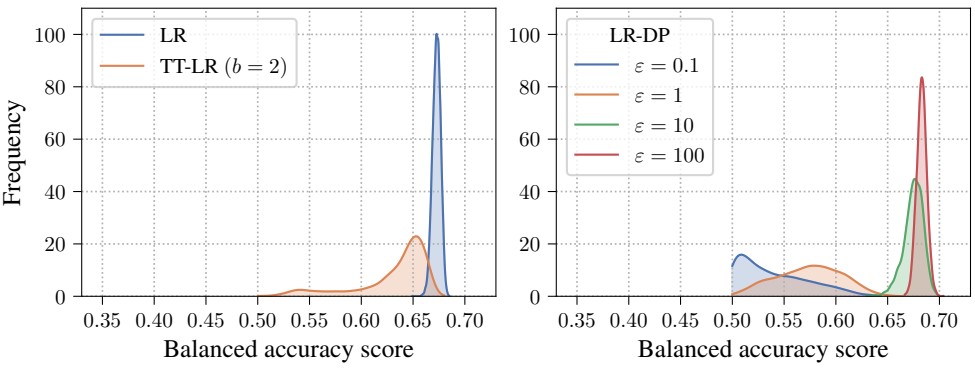

Figure 3: Balanced accuracy distributions of models trained on Cho1, evaluated on all cohorts. Left panel: LR vanilla and TT-LR with $b = 2$. Right panel: LR-DP models across privacy budgets $\varepsilon$, showing how accuracy improves as $\varepsilon$ increases and injected noise decreases. Degenerate TT runs with accuracy near 50% are rare and do not affect median values.

Table 5 reports median balanced accuracies and AUC scores across all cohorts for models trained on Cho1. Note that balanced accuracies use a threshold based on Youden's J statistic rather than the standard 0.5 threshold, which produced irregular results for DP models under strong noise.

## A.2  Per-cohort attack accuracies

Table 6 reports per-cohort Hamming scores for LR, NN, and TT models, i.e., the proportion of correct membership predictions for each cohort individually. These results show that privacy vulnerabilities extend to identifying all cohorts with high confidence regardless of their size, although the highest scores are generally obtained for the largest cohorts, which are the easiest to identify.

Table 5: Median balanced accuracies (left block) and median AUC scores (right block) of models trained on Cho1, evaluated on each cohort. Balanced accuracies use a threshold based on Youden's J statistic. These results complement the attack scores reported in the main text by showing the utility cost associated with each privacy mechanism under the same setting as LORIS.

| | | Balanced accuracy | | | | | | AUC | | | | | |
|---|---|---|---|---|---|---|---|---|---|---|---|---|---|
| | | Cho1 | Cho2 | MSK1 | MSK2 | Shim | Kato | Cho1 | Cho2 | MSK1 | MSK2 | Shim | Kato |
| LR | (vanilla) | **0.68** | **0.69** | **0.68** | **0.63** | **0.62** | **0.78** | **0.74** | **0.75** | **0.70** | **0.63** | **0.60** | **0.75** |
| | (averaged) | 0.68 | 0.69 | 0.69 | 0.63 | 0.62 | 0.78 | 0.74 | 0.75 | 0.70 | 0.63 | 0.60 | 0.71 |
| LR-DP | $(\varepsilon = 0.1)$ | 0.52 | 0.53 | 0.53 | 0.56 | 0.53 | 0.58 | 0.49 | 0.49 | 0.50 | 0.49 | 0.49 | 0.49 |
| | $(\varepsilon = 1)$ | 0.56 | 0.57 | 0.56 | 0.60 | 0.56 | 0.62 | 0.56 | 0.57 | 0.55 | 0.55 | 0.54 | 0.51 |
| | $(\varepsilon = 10)$ | **0.67** | **0.68** | **0.66** | **0.63** | **0.61** | **0.70** | **0.72** | **0.73** | **0.68** | **0.62** | **0.59** | **0.64** |
| | $(\varepsilon = 100)$ | 0.68 | 0.69 | 0.68 | 0.63 | 0.62 | 0.78 | 0.74 | 0.75 | 0.70 | 0.63 | 0.60 | 0.75 |
| TT-LR | $(b = 2)$ | **0.66** | **0.66** | **0.65** | **0.63** | **0.61** | **0.70** | **0.69** | **0.69** | **0.67** | **0.62** | **0.60** | **0.62** |
| | $(b = 6)$ | 0.67 | 0.67 | 0.67 | 0.63 | 0.62 | 0.72 | 0.72 | 0.72 | 0.69 | 0.63 | 0.60 | 0.65 |
| | $(b = 10)$ | 0.67 | 0.68 | 0.67 | 0.63 | 0.62 | 0.73 | 0.72 | 0.72 | 0.69 | 0.63 | 0.60 | 0.65 |
| NN | | **0.72** | **0.69** | **0.64** | **0.63** | **0.62** | **0.80** | **0.78** | **0.74** | **0.66** | **0.61** | **0.63** | **0.75** |
| NN-DP | $(\varepsilon \approx 0.2)$ | 0.58 | 0.59 | 0.53 | 0.63 | 0.57 | 0.62 | 0.59 | 0.59 | 0.49 | 0.60 | 0.53 | 0.55 |
| | $(\varepsilon \approx 1)$ | 0.60 | 0.61 | 0.54 | 0.64 | 0.59 | 0.63 | 0.61 | 0.61 | 0.52 | 0.62 | 0.55 | 0.55 |
| | $(\varepsilon \approx 10)$ | 0.61 | 0.62 | 0.56 | 0.64 | 0.60 | 0.65 | 0.65 | 0.63 | 0.56 | 0.64 | 0.57 | 0.57 |
| | $(\varepsilon = \infty)$ | **0.68** | **0.68** | **0.63** | **0.66** | **0.62** | **0.75** | **0.73** | **0.72** | **0.65** | **0.63** | **0.62** | **0.73** |
| TT-NN | $(b = 2)$ | **0.66** | **0.67** | **0.63** | **0.65** | **0.61** | **0.72** | **0.70** | **0.69** | **0.65** | **0.62** | **0.60** | **0.67** |
| | $(b = 6)$ | 0.68 | 0.69 | 0.65 | 0.64 | 0.62 | 0.77 | 0.73 | 0.74 | 0.67 | 0.62 | 0.62 | 0.73 |
| | $(b = 10)$ | 0.68 | 0.69 | 0.65 | 0.64 | 0.62 | 0.75 | 0.73 | 0.73 | 0.67 | 0.62 | 0.62 | 0.71 |

Table 6: Hamming scores of adversarial classifiers on vanilla models, reported separately for each cohort. A score of 0.5 corresponds to random guessing and 1.0 to perfect identification. Standard deviations are generally of order $10^{-3}$–$10^{-2}$ and do not exhibit systematic differences across settings; hence, we omit them for readability. Results complement the aggregate scores reported in the main text by showing which cohorts are most easily identified.

| | | Cho1 | Cho2 | MSK1 | MSK2 | Shim | Kato |
|---|---|---|---|---|---|---|---|
| LR (vanilla) | bBB | 0.9412 | 0.9358 | 0.9089 | 0.7237 | 0.7713 | 0.6259 |
| | cBB | 0.9945 | 0.9782 | 0.9616 | 0.9626 | 0.9130 | 0.6675 |
| | WB | 0.9982 | 0.9915 | 0.9678 | 0.9716 | 0.9152 | 0.7537 |
| TT-LR $(b = 2)$ | bBB | 0.7530 | 0.6927 | 0.7100 | 0.6614 | 0.6357 | 0.5471 |
| | cBB | 0.9343 | 0.8891 | 0.8740 | 0.8950 | 0.7517 | 0.5943 |
| | WB$^\star$ | 0.8671 | 0.8064 | 0.7820 | 0.8517 | 0.5975 | 0.5722 |
| NN | bBB | 0.9256 | 0.8826 | 0.7948 | 0.6104 | 0.6788 | 0.5329 |
| | cBB | 0.9964 | 0.9910 | 0.9907 | 0.7763 | 0.8552 | 0.5552 |
| | WB | 0.8335 | 0.7370 | 0.6545 | 0.5558 | 0.5111 | 0.5098 |
| TT-NN $(b = 2)$ | bBB | 0.6429 | 0.6347 | 0.5832 | 0.5449 | 0.5450 | 0.5049 |
| | cBB | 0.7832 | 0.6859 | 0.6715 | 0.5367 | 0.5305 | 0.5023 |
| | WB | 0.5140 | 0.5133 | 0.5071 | 0.4972 | 0.5002 | 0.5049 |

### A.3 Performance of models trained on Cho1 vs. Cho1+Kato

Table 7 reports median balanced accuracies and AUC scores of models trained on Cho1 alone or on Cho1+Kato, evaluated on Kato. These results provide context for the attack scores discussed in the main text: TT models show some performance degradation relative to LR, especially in AUC, but this alone does not explain the attack differences, as NNs achieve higher accuracies while leaking no information.

Table 7: Median balanced accuracies and AUC scores of models trained on Cho1 or Cho1+Kato, evaluated on Kato (35 patients). These results support the attack findings in the main text by showing that performance differences between the two training conditions are similar across model types, and cannot alone explain the large differences in attack success.

| | | Bal. accuracy | | AUC | |
|---|---|---|---|---|---|
| | | Cho1 | Cho1+Kato | Cho1 | Cho1+Kato |
| LR | (vanilla) | 0.7833 | 0.8000 | 0.7533 | 0.7733 |
| | (averaged) | 0.7833 | 0.8167 | 0.7133 | 0.7800 |
| TT-LR | $(b = 2)$ | 0.7167 | 0.7500 | 0.6167 | 0.6667 |
| NN | | 0.8000 | 0.8167 | 0.7467 | 0.8067 |
| TT-NN | $(b = 2)$ | 0.7167 | 0.7333 | 0.6733 | 0.6733 |

## B Efficient computations with TTs

A major advantage of tensor networks is their ability to represent high-order tensors using only a polynomial number of parameters. The TT representation of a tensor $T$ given by Eq. equation 2 requires only $\mathcal{O}(Ndr^2)$ coefficients when all cores $G_n$ are $r \times r$ matrices, as opposed to the $d^N$ coefficients needed for a general tensor $T \in \mathbb{R}^{d^N}$. While compactness does not automatically imply fast computation, TTs are efficient to evaluate: computing $T(i_1, \ldots, i_N)$ scales polynomially in $N$, unlike higher-dimensional TNs where evaluation may require exponential time.

Beyond evaluating samples, TTs enable efficient marginalization. Suppose $T$ encodes a probability distribution via the Born rule, $p(i_1, \ldots, i_N) = |T(i_1, \ldots, i_N)|^2$. Computing the partition function,

$$Z = \sum_{i_1,\ldots,i_N} p(i_1, \ldots, i_N), \tag{4}$$

is generally exponential in $N$, but in TT form it reduces to polynomial time by contracting each core with itself:

$$H_n(\alpha_{n-1}, \beta_{n-1}, \alpha_n, \beta_n) = \sum_{i_n} G_n(\alpha_{n-1}, i_n, \alpha_n) \, G_n(\beta_{n-1}, i_n, \beta_n), \tag{5}$$

yielding $r^2 \times r^2$ matrices $H_n$. Multiplying all $H_n$ sequentially produces $Z$ efficiently.

A similar procedure yields marginals by contracting only the cores of marginalized features. For instance, for a 2-site TT

$$T(i, j) = G_1(i)G_2(j), \tag{6}$$

the marginal $p(i)$ is

$$p(i) = \sum_{\alpha,\beta} G_1(i,\alpha) G_1(i,\beta) \, H_2(\alpha,\beta), \tag{7}$$

showing that marginals correspond to duplicate TTs with some cores contracted.

TT representations also enable efficient computation of conditional models without retraining. To compute $p(i_1, \ldots, i_{n-1}, i_{n+1}, \ldots, i_N \mid i_n = \mathbf{i}_n)$, it suffices to absorb the fixed feature into its neighbor:

$$\widetilde{G}_{n-1}(i_{n-1}) = G_{n-1}(i_{n-1}) \, G_n(\mathbf{i}_n), \tag{8}$$

which defines a reduced, conditioned TT

$$\widetilde{T}(i_1, \ldots, i_{n-1}, i_{n+1}, \ldots, i_N) = G_1(i_1) \cdots \widetilde{G}_{n-1}(i_{n-1}) G_{n+1}(i_{n+1}) \cdots G_N(i_N). \tag{9}$$

For further details on TTs and related tensor networks, see Cirac et al. (2021).

## C  Data standardization and parameter rescaling

Before training on each dataset $D = \{(x_1^k, \ldots, x_n^k, y^k)\}_k$, input features $x_1, \ldots, x_n$ are standardized as

$$\tilde{x}_j^k = \frac{x_j^k - \mu_j}{\sigma_j}, \tag{10}$$

where $\mu_j$ and $\sigma_j$ denote the mean and standard deviation of feature $j$, respectively.

LR models are trained on these standardized inputs, but their parameters must be corrected in order to operate directly on raw features. Let $\tilde{\theta} = (\tilde{\mathbf{w}}, \tilde{b})$ be the parameters obtained after training, defining

$$\Phi_{\tilde{\theta}}(\mathbf{x}) = \text{sigmoid}(\tilde{\mathbf{w}}^\mathsf{T} \mathbf{x} + \tilde{b}), \quad \text{where} \quad \text{sigmoid}(z) = \frac{1}{1 + e^{-z}}. \tag{11}$$

The corrected parameters are $\theta = (\mathbf{w}, b)$ with

$$w_j = \frac{\tilde{w}_j}{\sigma_j}, \quad b = \tilde{b} - \sum_j \frac{\tilde{w}_j \mu_j}{\sigma_j}. \tag{12}$$

This transformation ensures that trained models can be applied directly to raw inputs without explicit feature standardization.

An analogous rescaling applies to TT models. Consider a tensorized model with parameters $\widetilde{W}$,

$$\tilde{f}(x_1, \ldots, x_N) = \sum_{i_1, \ldots, i_N} \widetilde{W}(i_1, \ldots, i_N) \, \phi_1^{i_1}(x_1) \cdots \phi_N^{i_N}(x_N), \tag{13}$$

where $\phi_n(x) = [1, x]$ are polynomial embeddings (input dimension $d = 2$), and

$$\widetilde{W}(i_1, \ldots, i_N) = \widetilde{G}_1(i_1) \cdots \widetilde{G}_N(i_N). \tag{14}$$

To compensate for feature standardization, we define a new coefficient tensor W from corrected cores $G_n$ such that

$$G_n(1) = \widetilde{G}_n(1) - \frac{\mu_j}{\sigma_j} \widetilde{G}_n(2),$$
$$G_n(2) = \frac{1}{\sigma_j} \widetilde{G}_n(2).$$

(15)

The resulting TT parameters are thus expressed in terms of raw input features, analogous to the LR case.

## D    Recovering LR coefficients from cBB access

Since logistic regression is linear in the log-odds space,

$$\text{logit}(\mathbf{x}) = \log \frac{p(y = 1 \mid \mathbf{x})}{p(y = 0 \mid \mathbf{x})} = \mathbf{w}^{\mathsf{T}}\mathbf{x} + b,$$

(16)

its parameters can be exactly recovered from model evaluations on carefully chosen inputs. If queries to the zero vector and one-hot vectors $\mathbf{e}_j$ are allowed, the intercept $b$ is simply the logit at the zero vector, and each coefficient $w_j$ is given by the difference between the logit at $\mathbf{e}_j$ and the intercept.

More generally, when queries are restricted to inputs with all features strictly positive (as is the case when attacking LORIS through its web interface), $w_j$ can be recovered from two inputs $\mathbf{x}, \mathbf{x}'$ that differ only in feature $j$:

$$w_j = \frac{\text{logit}(\mathbf{x}) - \text{logit}(\mathbf{x}')}{x_j - x'_j}.$$

(17)

Once the weights are obtained, the intercept can be recovered from

$$b = \text{logit}(\mathbf{x}) - \mathbf{w}^{\mathsf{T}}\mathbf{x}$$

(18)

for any input $\mathbf{x}$.

Exact recovery assumes sufficiently precise continuous scores and the ability to issue suitable queries. Output rounding or discretization, restrictions on the query domain, and TT approximation errors can all affect reconstruction accuracy. Deriving a quantitative bound would require accounting for all these factors. We leave this analysis for future work.

