# OpenReview forum: "Private and interpretable clinical prediction with quantum-inspired tensor train models"
_TMLR — Under review for TMLR_

### Review · Reviewer_FEQi · 2026-06-05

**Summary Of Contributions:**

This paper studies privacy risks in publicly deployed clinical machine learning models and proposes a post-hoc defense based on tensor-train (TT) representations. The work combines a realistic attack framework, a practical defense, and interpretability analysis.

Attack. The authors develop cohort-level membership inference attacks under three adversarial access models: binary black-box (bBB), continuous black-box (cBB), and white-box (WB). As a central case study, they attack LORIS, a publicly available logistic regression model for immunotherapy response prediction hosted on a government website. They show that model parameters can be recovered from the interface and used to identify the training cohort with near certainty. More generally, they demonstrate that cohort membership, including small sub-cohorts, can be reliably inferred, and that repeated cross-validation with parameter averaging significantly amplifies privacy leakage.

Defense. To mitigate these risks, the paper proposes a post-hoc approach: discretize model outputs into $b$ bins and construct a tensor-train approximation using TT-RSS on a small pivot set. The resulting representation obfuscates original parameters via gauge transformations (eliminating white-box leakage) and empirically reduces black-box attack success rates. The defense requires no retraining and can be applied to existing deployed models.

Interpretability. The TT representation preserves key interpretability properties. For logistic regression, feature sensitivities derived via TT marginalization align with original coefficients, conditional analyses can be performed without retraining, and monotonicity is preserved. The framework also extends interpretability tools to neural networks.

Strengths:
- Addresses a practically important and underexplored privacy risk in clinical ML deployment.
- Includes a compelling real-world case study on a live public model (LORIS).
- Introduces a cohort-level privacy perspective beyond standard individual membership inference.
- Identifies the important and counterintuitive result that cross-validation averaging amplifies leakage.
- Proposes a practical post-hoc defense that does not require retraining or access to original data.
- Combines privacy mitigation and interpretability in a unified framework.
- Provides extensive empirical evaluation across models, access settings, and cohorts.
- Generally clear, well-structured, and accessible.

Weaknesses:
- Black-box privacy protection is empirical and attack-dependent, but is sometimes presented too close to formal guarantees such as Differential Privacy (DP).
- The contribution of discretization versus tensorization is not fully disentangled.
- Limited evaluation on small models (logistic regression and shallow neural networks); scalability to larger architectures is unclear.
- Interpretability claims are partly qualitative, especially for neural networks.
- Threat model focuses on cohort-level inference and does not address individual-level privacy.
- Occasional degenerate TT runs are not analyzed.

**Additional Comments:**

I found the paper well motivated, technically sound, and practically relevant. The LORIS case study and the cross-validation leakage finding are particularly strong contributions.

My main concern is conceptual: the manuscript occasionally places empirical TT-based protection too close to formally guaranteed privacy frameworks such as Differential Privacy. This should be clarified to avoid overstating guarantees.

The role of discretization versus tensorization also remains somewhat unclear and deserves further investigation.

Subject to addressing the key issues, particularly clarifying the nature of privacy guarantees, analyzing degenerate cases, and refining the threat model. I am positive about the paper.

Overall recommendation: Leaning Accept.

**Audience:**

Yes

**Audience Explanation:**

The paper lies at the intersection of several active and relevant areas: machine learning privacy, clinical AI, membership inference attacks, and tensor methods.

- Privacy and security researchers will find the cohort-level attack formulation and leakage amplification via cross-validation particularly valuable.
- Clinical ML practitioners will benefit from concrete insights into risks associated with deploying models via public interfaces.
- Researchers in tensor methods and interpretable ML may find the use of TT representations as a post-hoc privacy and interpretability tool novel and interesting.

The cross-validation leakage result is especially impactful, as it challenges a widely used best practice and has immediate implications beyond the specific application domain.

Overall, the paper offers insights of broad relevance even outside immunotherapy prediction.

**Broader Impact Concerns:**

The paper includes an appropriate broader-impact discussion and acknowledges the dual-use nature of the work.

The demonstrated attacks could be used to infer participation of cohorts in deployed clinical models. However, the work is primarily defensive, proposes mitigation strategies, and relies on publicly available data and models.

Two points should be emphasized further:

- Responsible disclosure regarding the attacked live system.

- Clear communication that TT-based protection does not replace formal privacy mechanisms such as Differential Privacy, particularly for individual-level guarantees.



These concerns do not prevent publication but should be addressed to avoid misinterpretation or misuse.

**Claims And Evidence:**

Yes

**Claims Explanation:**

The paper provides strong empirical evidence supporting its main claims.

- The LORIS case study is particularly compelling: parameter recovery and cohort identification are demonstrated concretely, with near-perfect attack success (e.g., scores close to $1.0$).
- The general attack framework is well validated using standard shadow-model and meta-classifier methodology, evaluated across multiple access settings and model classes.
- The cross-validation leakage finding is clearly supported: averaged models exhibit substantially higher attack success rates while maintaining similar predictive performance.
- The TT defense convincingly eliminates white-box leakage due to gauge freedom, with empirical results showing near-random attack success ($\sim 0.5$).

However, an important limitation concerns black-box privacy claims. The paper presents TT-based protection as ``comparable'' to Differential Privacy in some places, but this comparison is purely empirical, based on attack success rates under specific settings, without formal guarantees such as $(\epsilon, \delta)$-DP. This distinction is not always clearly maintained, especially in high-level descriptions.

Additionally, the current experiments do not fully separate the effects of discretization and tensorization, making attribution of gains less precise.

These issues do not invalidate the results, but they limit the strength and generality of the conclusions.

**Requested Changes:**

Critical for my recommendation:

- Clearly and consistently state that TT-based protection provides empirical privacy against evaluated attacks, not formal guarantees. Comparisons with Differential Privacy must be carefully qualified to avoid suggesting equivalence.
- Add an ablation study disentangling: (i) discretization alone, (ii) tensorization alone (if feasible), (iii) discretization followed by tensorization.
- Clarify the scope of the threat model, especially the distinction between cohort-level and individual-level privacy, and explicitly state limitations.
- Investigate and report the occurrence and causes of degenerate TT runs, including guidance for detection or mitigation.
- Expand discussion of threat-model assumptions, including adversary access to candidate cohorts.


Changes that would strengthen the paper but are not essential:

- Provide more quantitative evaluation of interpretability, especially for neural networks.
- Discuss scalability to higher-dimensional data and deeper architectures.
- Include analysis of pivot set size and its effect on performance and privacy.
- Clarify discretization procedures for multi-bin settings.
- Add clearer positioning and structured comparison with DP baselines.
- Include a brief responsible disclosure statement regarding the LORIS case study.
- Expand discussion of limitations and potential failure modes.

---

> ### Author Response · Authors · 2026-07-13
> **Response to Reviewer FEQi**
>
> We thank the reviewer for the time and care devoted to this detailed and constructive review. Below, we address the reviewer’s concerns point by point.
>
> ---
> **Comparison with Differential Privacy**
>
> We agree that some of the original wording could suggest a closer equivalence between tensorization and DP than intended. DP is included as a practical baseline to contextualize the empirical privacy–utility trade-off, not as a formally equivalent mechanism.
>
> The wording has been revised in the **Abstract, Introduction, Sec. 3.1.2, Sec. 3.2.3, and Discussion**. The Introduction now explicitly states that *“This empirical comparison is not intended to imply equivalence…”*. Sec. 3.1.2 also clarifies that the experiments compare complete practical defense pipelines, since the available DP implementations require different optimization settings. The Discussion distinguishes the formal guarantees of DP from the attack-dependent BB protection of tensorization, while emphasizing its main practical advantage: it can be applied post-hoc without retraining or access to the original patient data. Combining tensorization with DP-style output perturbation is also discussed as future work.
>
> ---
> **Discretization and tensorization**
>
> We have further clarified the distinction between the WB protection provided by TT gauge freedom and the BB protection introduced through output discretization, as well as the role of $b$ throughout the manuscript.
>
> We did not add a tensorization-only baseline because similar experiments were reported in Pareja Monturiol et al. (2025), where TT-RSS was used to tensorize NNs and its effect on both WB and BB attacks was studied. Building on those results, the present work adds BB protection through discretization, since TT parameter obfuscation alone does not prevent attacks based on model evaluations.
>
> Table 3 also indicates the approach to the undiscretized case: for several models, predictive performance changes little between $b=6$ and $b=10$, suggesting that the larger-$b$ settings may already be close in utility to tensorizing the original continuous model, and therefore to the tensorization-only baseline.
>
> ---
> **Threat model and cohort-level inference**
>
> The distinction between cohort-level and individual-level inference was already explicit in the **Introduction and Sec. 2.1**, where the manuscript defines the attack as identifying which public patient cohorts were used for training and explains why this remains privacy-relevant despite being easier than individual membership inference. The attacker’s prior knowledge was also described in **Sec. 3.1.3**, including access to the candidate public cohorts and knowledge of the model architecture and approximate training procedure, with uncertainty over the specific cohorts and hyperparameters used.
>
> To reinforce this scope, we added two clarifications. First, **Sec. 3.1.3** now states that the adversary considers a known set of candidate public cohorts and does not attempt to discover arbitrary unknown cohorts. Second, the **Discussion** now explicitly limits the conclusions to the considered empirical threat model. The description of the 35-patient Kato experiment was also refined to clarify that it remains a group-level attack, while bringing the setting closer to individual membership inference than the broader cohort-level experiments.
>
> ---
> **Degenerate TT runs, pivots, and scalability**
>
> The discussion of degenerate runs has been expanded in **Secs. 2.3 and 3.1.4**. The revised manuscript explains that TT-RSS relies on randomized linear algebra, that stability depends on the pivot set, and that increasing the pivot-set size can mitigate occasional near-chance approximations.
>
> Scalability is now described more cautiously. TT-RSS is architecture-agnostic because it only requires model evaluations, but practical applicability depends on whether the target function admits an efficient TT approximation. This limitation is stated in the **Introduction and Discussion**.
>
> ---
> **Interpretability for neural networks**
>
> Fig. 2a provides an initial quantitative comparison of TT-NN feature sensitivities with LR and TT-LR. The revised **Discussion** now explicitly acknowledges that the TT-NN interpretability analysis is limited and identifies broader mechanistic-interpretability studies as future work.
>
> ---
> **Responsible disclosure and broader impact**
>
> No formal disclosure process was conducted with the LORIS authors. However, all analyses used only publicly released interfaces, parameters, source code, and cohorts, and no non-public patient data were accessed or reconstructed.
>
> This has been clarified in the revised **Broader Impact Statement**.
>
> ---
> We thank the reviewer again for the detailed feedback, which helped improve the precision of the manuscript regarding the privacy claims, threat model, and roles of discretization and tensorization.

---

### Review · Reviewer_Nbze · 2026-06-15

**Summary Of Contributions:**

The paper makes an interesting and mostly well-supported empirical claim: publicly released clinical prediction models can leak training-cohort membership, and tensor-train tensorization of discretized model outputs can reduce this leakage while preserving reasonable utility and interpretability. The strongest evidence is the LORIS case study, where the authors identify the Cho1 training cohort from released coefficients and from coefficients reconstructed through the public interface, plus broader shadow-model experiments across LR, NN, DP, and TT variants.

**Audience:**

Yes

**Audience Explanation:**

The LORIS case study gives the work practical relevance because it concerns a real publicly available clinical prediction model rather than only a synthetic benchmark. The finding that repeated cross-validation/averaging can increase LR membership leakage is particularly noteworthy, since cross-validation is commonly used to stabilize clinical models and is usually viewed as a benign or beneficial practice.

**Claims And Evidence:**

Yes

**Claims Explanation:**

The paper provides a reasonably convincing empirical case that clinical prediction models, especially logistic-regression models such as LORIS, can leak cohort-membership information.
However, I would ask the authors to temper several claims. First, statements such as “identify the training cohort with certainty” and “fully obfuscates model parameters” are stronger than the evidence warrants. The former is based on a meta-classifier score on a single known LORIS target, not a formal guarantee of certainty. The latter relies on gauge randomization and previous tensor-network privacy arguments, but the paper does not provide a formal privacy guarantee for the full TT-RSS-plus-discretization pipeline. Second, the comparison with differential privacy is only empirical and attack-specific. DP provides a formal individual-level privacy guarantee. Finally, there appears to be at least one clarity or consistency issue: the aggregate NN cBB score in Table 3 is very high, while the per-cohort NN cBB scores in Table 6 appear much lower on average.

**Requested Changes:**

The current wording sometimes goes beyond what is demonstrated empirically. The results support empirical attack resistance under the authors’ evaluated cohort-level threat model, not a formal privacy guarantee comparable to DP.

The authors should describe it as high-confidence identification under their trained attack, report uncertainty across attack repetitions if available, and clearly separate “known actual training cohort” from “predicted cohort membership score.”

The authors should either add controls that isolate the effect of DP noise from implementation changes, or explicitly state that these are practical DP baselines rather than controlled comparisons. Claims such as “comparable to DP” should be rewritten as “comparable empirical attack resistance in this experimental setup.”

---

> ### Author Response · Authors · 2026-07-13
> **Response to Reviewer Nbze**
>
> We thank the reviewer for the careful assessment of the manuscript and, in particular, for identifying the inconsistencies in the reported tables. Below, we address the reviewer’s concerns point by point.
>
> ---
> **Scope and wording of the privacy claims**
>
> We agree that expressions such as *“identify the training cohort with certainty”* and *“fully obfuscates model parameters”* were too strong without further qualification. The wording has been revised throughout the **Abstract, Introduction, Secs. 2.3 and 3.2.1, and Discussion**.
>
> The LORIS result is now described as identification *“with high confidence”*. Sec. 3.2.1 also separates the known ground truth (Cho1 is reported as the training set in Chang et al. (2024)) from the membership probabilities produced by the meta-classifier.
>
> The discussion of TT parameter obfuscation now clarifies that the formal WB guarantee concerns information beyond the represented BB behavior. Accordingly, the empirical result is stated more directly: attacks applied to the gauge-randomized TT parameters achieve scores close to random guessing, while information retained in the represented function may still be accessible through BB attacks.
>
> ---
> **Comparison with Differential Privacy**
>
> We agree that the original wording could suggest formal equivalence between tensorization and DP. The comparison is intended only to place the empirical attack scores and predictive utility of the TT models alongside practical DP baselines.
>
> This distinction is now explicit in the **Abstract, Introduction, Sec. 3.1.2, Sec. 3.2.3, and Discussion**. The Introduction states that the comparison *“is not intended to imply equivalence…”*, while Sec. 3.1.2 clarifies that the available implementations use different optimization settings and therefore represent complete practical defense pipelines rather than a controlled comparison of noise alone. The Discussion further distinguishes formal DP guarantees from the attack-dependent BB protection observed for tensorization.
>
> ---
> **Corrections to Tables 3, 4, and 6**
>
> We thank the reviewer for noticing the discrepancy between the aggregate NN cBB result in Table 3 and the per-cohort values in Table 6. Rechecking the full pipeline revealed two reporting errors.
>
> First, Table 3 mistakenly contained the maxima across five repetitions rather than their means. Second, the cBB attacks for NN and TT-NN in Tables 3 and 4 were still using binary outputs instead of continuous scores.
>
> These results have been recomputed and corrected in **Tables 3, 4, and 6**. The aggregate scores in Table 3 are now consistent with the per-cohort results in Table 6. The corrections slightly modify several values but do not change the qualitative conclusions.
>
> ---
> **Uncertainty across repetitions**
>
> We agree that the variability across attack repetitions should be reported. The Hamming scores correspond to means over five repetitions, and this is now stated in **Sec. 3.1.3** and the relevant table captions.
>
> The standard deviations are generally of order $10^{-3}$–$10^{-2}$ and show no systematic differences across settings. To avoid substantially enlarging an already dense table, these values are summarized in the text and captions rather than reported in every cell.
>
> ---
> **Interpretation of the LORIS attack**
>
> The revised **Sec. 3.2.1** now makes explicit that Cho1 is independently known to be the LORIS training set from Chang et al. (2024). Table 2 reports the probabilities assigned by the attack meta-classifier and is evaluated against this known ground truth.
>
> The statement that Cho1 is identified with high confidence therefore refers to the meta-classifier assigning its dominant score to the correct cohort, rather than to a formal certainty guarantee.
>
> ---
> We thank the reviewer again for the careful reading, which led both to clearer privacy claims and to the correction of the reported results.

---

### Review · Reviewer_6tx5 · 2026-06-29

**Summary Of Contributions:**

The authors developed cohort-level membership inference attacks under three adversarial access levels (binary black-box, continuous black-box, and white-box). They successfully demonstrated that public clinical models like LORIS leak training cohort information. They introduced a quantum-inspired defense that transforms discretized model outputs into tensor trains (TTs) via TT-RSS, achieving parameter obfuscation and output space compression. They showed that TT representations preserve clinical interpretability and extend it to black-box neural networks (NNs) through efficient marginal and conditional distribution computations.

**Audience:**

Yes

**Audience Explanation:**

This paper directly addresses two core, high-interest themes within the TMLR community: privacy-preserving ML and explainable AI. By demonstrating that common practices like cross-validation introduce hidden security risks, and by introducing a novel, quantum-inspired tensor train framework that bridges the gap between strict black-box obfuscation and clinical interpretability, the findings provide actionable insights and foundational methodologies that directly appeal to machine learning theoreticians and applied clinical practitioners alike.

**Broader Impact Concerns:**

Although the authors propose and evaluate a post-hoc defense mechanism based on tensor networks to reduce information exposure, it does not completely eliminate all vulnerabilities. Experiments demonstrate that, for tensorized logistic regression (TT-LR) models, an attacker can still approximately reconstruct the original coefficients through model output evaluations, thereby maintaining a high success rate under white-box attacks. This indicates that the defense remains vulnerable to adversaries equipped with specific domain expertise.

**Claims And Evidence:**

Yes

**Claims Explanation:**

The submission provides exceptionally clear and convincing empirical evidence to back its core claims. By launching three meticulously designed membership inference attacks against LORIS, a publicly deployed clinical model, the authors definitively demonstrate real-world privacy risks by successfully recovering parameters and identifying training cohorts. To support their quantum-inspired tensor train defense, they present structured evaluations showing that gauge transformations successfully reduce white-box attacks to random guessing while maintaining high predictive accuracy. Furthermore, the evidence clearly shows that the TT representation preserves score monotonicity and enables advanced, subgroup-specific feature sensitivity analysis without retraining, validating its dual utility for privacy and interpretability.

**Requested Changes:**

The empirical comparison between the Tensor Train defense and Differential Privacy lacks theoretical depth. Please provide formal privacy bounds mathematically linking the discretization parameter $b$ to the $\epsilon$-DP budget, potentially via output perturbation, to strengthen the framework's foundation.
The results show attackers can reconstruct original logistic regression coefficients from TT evaluations, maintaining high white-box attack success. Please propose a concrete mitigation strategy for this vulnerability or provide a theoretical lower bound on the reconstruction error.

---

> ### Author Response · Authors · 2026-07-13
> **Response to Reviewer 6tx5**
>
> We thank the reviewer for the careful assessment of the manuscript and for the specific suggestions concerning formal privacy guarantees and the LR reconstruction attack. Below, we address these concerns point by point.
>
> ---
> **Comparison with Differential Privacy and the role of $b$**
>
> We agree that the original wording could suggest a closer relationship between tensorization and DP than intended. The comparison is empirical and is included to contextualize the observed privacy–utility trade-off against practical DP baselines, not to claim formal equivalence.
>
> This has been clarified in the **Abstract, Introduction, Sec. 3.1.2, Sec. 3.2.3, and Discussion**. In particular, the revised text distinguishes the formal guarantees of DP from the attack-dependent BB protection observed for tensorization and notes that the evaluated methods correspond to complete practical pipelines with different optimization requirements.
>
> We have also clarified that $b$ and $\varepsilon$ control fundamentally different quantities. The parameter $b$ determines how much output information from a fixed trained model is retained before tensorization, whereas $\varepsilon$ bounds the behavior of a randomized mechanism under changes to the training dataset. Therefore, $b$ alone cannot define an equivalent $\varepsilon$ without additional assumptions on model sensitivity, randomization, and the query mechanism.
>
> Accordingly, Sec. 3.2.3 now presents $b=2$ only as a soft empirical upper bound on the information recoverable while preserving the original binary classification behavior. Deriving formal information bounds for general $b$, or replacing discretization with a calibrated DP output mechanism, is left for future work. The Discussion also emphasizes that tensorization is post-hoc and can be combined with DP-style output perturbation to introduce formal output-level guarantees.
>
> ---
> **White-box guarantees and LR coefficient reconstruction**
>
> The revised manuscript further clarifies that the formal TT guarantee concerns parameter-level information: gauge freedom ensures that access to the TT cores reveals no information beyond the represented BB function. It does not prevent an attacker from exploiting information that remains accessible through model evaluations.
>
> The starred TT-LR attacks do not recover LR coefficients directly from the randomized TT cores. Instead, they evaluate the TT and, assuming knowledge of the underlying LR architecture, reconstruct coefficients from its approximately logistic input-output behavior. This distinction is now explained in **Secs. 2.3, 3.1.3, and 3.2.3**.
>
> The revised text also stresses that this is an architecture-specific attack requiring prior knowledge that the TT approximates an LR, and that no analogous recovery procedure is available for the TT-NN experiments.
>
> ---
> **Mitigating the LR-specific attack**
>
> The most direct mitigation is to strengthen protection of the represented BB function. The revised manuscript therefore discusses replacing deterministic discretization with randomized output perturbation, potentially calibrated to provide DP guarantees, before tensorization.
>
> The attack also relies on sufficiently precise continuous scores, suitable queries, and knowledge of the underlying LR form. These assumptions, together with the role of output rounding, query restrictions, and TT approximation error, are now summarized in **Sec. 3.1.3 and Appendix D**. A quantitative reconstruction-error bound is left for future work.
>
> ---
> **Scope of the privacy claims**
>
> The **Discussion** now explicitly limits the conclusions to the considered cohort-level attacks over a known set of public cohorts and states that the observed BB protection is empirical rather than a formal privacy guarantee.
>
> ---
> We thank the reviewer again for these suggestions, which helped clarify the distinction between formal WB protection, empirical BB protection, and DP, as well as the assumptions and possible mitigations of the LR-specific reconstruction attack.

---

> > ### Comment · Reviewer_6tx5 · 2026-07-14
> >
> > Thank the authors for their detailed response and the extensive revisions made to the manuscript. My previous concerns regarding the formal privacy guarantees and the LR reconstruction attack have been well addressed. However, a few remaining issues require your attention:
> >
> > **1. Defense Capabilities Against other Attacks**
> >
> > Although the proposed defense demonstrates effectiveness in mitigating the cohort-level membership inference attacks introduced in this work, the revised manuscript lacks a discussion on the defense mechanism's effectiveness against other prominent attacks not evaluated in this study, such as model inversion attacks (Fredrikson et al., 2014) and classical individual-level membership inference attacks (Shokri et al., 2017). The proposed method might provide insufficient protection against these specific approaches, and this limitation should be explicitly discussed in the text.
> >
> > **2. Theoretical Basis for Pivot Set Size ($\vert{}D\vert{}$)**
> >
> > In Sections 2.3 and 3.1.4, the manuscript notes that the TT-RSS algorithm occasionally generates degenerate Tensor Train (TT) models with near 50% accuracy, attributing this to "intrinsic randomness." You state that increasing the pivot dataset size ($\vert{}D\vert{}$) mitigates this issue, but this claim relies solely on empirical observations from Pareja Monturiol et al. (2025). Please provide support in Section 2.3 or the Appendix. Specifically, include a sample complexity analysis rooted in randomized linear algebra, or at least mathematical intuition establishing a theoretical lower bound relationship between $\vert{}D\vert{}$ and the model hyperparameters ($N, d, r$) to ensure decomposition stability.
> >
> > **3. Missing Discretization Baseline**
> >
> > Echoing the concerns raised by Reviewer FEQi regarding the ablation studies, the justification for omitting the "discretization-only" experiment remains insufficient. Although the text attempts to reiterate that white-box security relies on tensorization and black-box security on discretization, the revised statements in Section 2.3 indicate that this theoretical separation is not entirely robust. While the authors have clarified why a tensorization baseline is structurally challenging to evaluate under certain access levels, the omission of a pure discretization only control group remains unjustified. Without a "discretization-only" control group, reviewers cannot accurately judge the actual contribution of the complex TT-RSS mechanism in defending against black-box attacks.

---

> > > ### Author Response · Authors · 2026-07-15
> > > **Response to Reviewer 6tx5**
> > >
> > > We thank the reviewer for the thoughtful follow-up and for acknowledging that the previous concerns were addressed. We have not further revised the manuscript, but clarify below how these points relate to the scope and design of the work.
> > >
> > > **1. Defense capabilities against other attacks**
> > >
> > > We agree that the experiments do not establish protection against model inversion or individual-level membership inference. As explicitly stated in the revised **Discussion**, our conclusions are limited to the attacks considered in this work.
> > >
> > > Related output-obfuscation methods have shown that transforming or perturbing confidence scores can reduce both membership inference and model inversion attacks (Yang et al., 2020; 2023; Ye et al., 2022). Individual membership inference targets a more granular signal and is generally harder than identifying a complete cohort under comparable assumptions. The 35-patient Kato experiment in **Sec. 3.2.4** approaches this smaller-group regime and yields similar conclusions with lower attack accuracies. Although it remains cohort-level, this suggests that attack accuracy may decrease further at the individual level; however, no such claim is made in the manuscript.
> > >
> > > As detailed below, **our core contribution is not discretization alone, but tensorization**, which is why the work focuses on the complete protected representation rather than testing discretization against a broad range of attacks.
> > >
> > > Moreover, the cohort setting reflects a realistic clinical scenario: candidate datasets may be associated with specific studies or institutions, and their contribution can be detected without exact knowledge of every sample. By contrast, some individual-level settings assume stronger side information, such as knowledge of the remaining $n-1$ training samples, which is less representative of the scenario considered here.
> > >
> > > **2. Theoretical basis for the pivot-set size**
> > >
> > > We agree that a theoretical sample-complexity characterization for TT-RSS would be highly relevant, but it remains open for general pretrained black-box models.
> > >
> > > Randomized linear algebra provides useful intuition: under leverage-score-type sampling, roughly $O(r\log r)$ samples can suffice to capture a rank-$r$ subspace, while uniform-sampling bounds additionally depend on coherence. However, these results assume sampling distributions aligned with the relevant matrix subspaces.
> > >
> > > Related TT methods do not provide the requested general result. Hur et al. (2023) obtain sketching guarantees only under specific structured settings, while tensor cross interpolation, as studied by Núñez Fernández et al. (2025), seeks an informative set of $r$ rows and columns using rank-revealing heuristics rather than a general sample-complexity bound.
> > >
> > > These ideas motivate drawing pivots from the training set, since the model is designed to represent its distribution, or the corresponding conditional distribution in classification. However, turning this intuition into rigorous bounds for our setting remains open. We therefore chose not to include arguments whose connection to TT-RSS is not yet established, and instead refer to Pareja Monturiol et al. (2025) for the empirical analysis and discussion.
> > >
> > > **3. Discretization-only baseline**
> > >
> > > We recognize that a discretization-only baseline could isolate the effect of output compression under a strictly enforced BB interface. The bBB and cBB attacks on the original models already illustrate the two extreme levels of output information considered here: binary classifications and full continuous scores, although not the intermediate cases.
> > >
> > > The central contribution, however, is not discretization as an external wrapper, but the **post-hoc tensorization of an output-obfuscated model**. A discretization-only wrapper provides no protection if the original parameters or raw continuous outputs remain accessible or the wrapper can be bypassed.
> > >
> > > Tensorization instead constructs a new deployable function using only the obfuscated outputs for its construction. The original parameters and continuous behavior are no longer released, and TT gauge freedom ensures that even access to the TT parameters reveals only information contained in the resulting BB function. This also applies to the starred TT-LR attack, which reconstructs coefficients from TT evaluations and therefore uses only BB information. Thus, the role of tensorization is to create an interpretable model that intrinsically preserves only that restricted BB information while extending the protection to WB access.
> > >
> > > For this reason, and as already discussed in **Fig. 1, Secs. 2.2–2.3, Sec. 3.1.2, and the Discussion**, the framework is also not tied to discretization: other output-obfuscation mechanisms, including methods with formal DP guarantees, could be applied before tensorization.

---

### Author Response · Authors · 2026-07-13
**Summary of Revisions and Responses to the Reviewers**

We thank all reviewers for their careful reading and valuable feedback. The manuscript has been revised substantially to improve the precision of the privacy claims, clarify the experimental setting, and correct the reported results. The main changes are summarized below:

- **Comparison with Differential Privacy:** DP is now presented as a practical baseline for contextualizing the empirical privacy–utility trade-off, without implying formal equivalence with tensorization.

- **Discretization and tensorization:** Their roles have been more clearly separated. Output discretization provides empirical BB protection by restricting the information retained from the original model, while TT gauge freedom provides WB protection by obfuscating parameter-level information beyond the represented BB behavior.

- **Scope of the TT guarantees:** The manuscript now clarifies that WB protection does not prevent attacks based on information that remains accessible through model evaluations.

- **Threat model:** The adversary’s access to a known set of candidate public cohorts is now stated more explicitly, together with the distinction between cohort-level and individual-level membership inference.

- **Role of $b$:** The relation between $b$ and $\varepsilon$-DP has been clarified. The revised manuscript no longer suggests a direct correspondence and identifies formal information bounds or DP-style output perturbation as future work.

- **LR-specific reconstruction attack:** Its assumptions and limitations are now explained more clearly, including dependence on knowledge of the underlying architecture, sufficiently precise outputs, suitable queries, and TT approximation error.

- **TT limitations:** The discussion of degenerate approximations, pivot-set dependence, scalability, and the limited scope of the TT-NN interpretability analysis has been expanded.

- **Corrected results:** Tables 3, 4, and 6 have been corrected after rechecking the experimental pipeline. The aggregate scores now report means over repetitions, the NN and TT-NN cBB attacks use continuous outputs as intended, and uncertainty across repetitions is summarized in the text and captions.

- **Broader impact:** The revised statement clarifies that the analysis uses only publicly released interfaces, parameters, source code, and cohorts, without accessing or reconstructing non-public patient data.

We believe these revisions have made the scope, contributions, and limitations of the work substantially clearer, and we thank the reviewers again for helping improve the manuscript.